# Single-molecule kinetic locking allows fluorescence-free quantification of protein/nucleic-acid binding

Martin Rieu [1,2 ✉], Jessica Valle-Orero[1,2], Bertrand Ducos [1,2], Jean-François Allemand[1,2] & Vincent Croquette[1,2,3]

Fluorescence-free micro-manipulation of nucleic acids (NA) allows the functional characterization of DNA/RNA processing proteins, without the interference of labels, but currently fails to detect and quantify their binding. To overcome this limitation, we developed a method based on single-molecule force spectroscopy, called *kinetic locking*, that allows a direct in vitro visualization of protein binding while avoiding any kind of chemical disturbance of the protein's natural function. We validate *kinetic locking* by measuring accurately the hybridization energy of ultrashort nucleotides (5, 6, 7 bases) and use it to measure the dynamical interactions of *Escherichia coli/E. coli* RecQ helicase with its DNA substrate.

[1] Laboratoire de physique de l'Ecole Normale Supérieure (LPENS), ENS, Université PSL, CNRS, Sorbonne Université, Université de Paris, Paris, France. [2] Institut de Biologie de l'Ecole Normale Supérieure (IBENS), ENS, Université PSL, CNRS, INSERM, Paris, France. [3] ESPCI Paris, Université PSL, 10 rue Vauquelin, 75005 Paris, France. ✉email: martin.rieu@ens.psl.eu

Single-molecule micromanipulation techniques such as atomic force microscope (AFM), optical tweezers, acoustic tweezers, or magnetic tweezers have extensively been used to study nucleic acids (NAs). The approach relies on stretching NAs by applying pico-Newton-scale forces and measuring their extension changes upon binding of the molecules of interest. While limited to in vitro measurements, they reveal features that are neither observable with bulk measurements nor with in cellulo fluorescence microscopy. For example, they unveiled the precise mechanisms of NA unwinding and replication by enzymes such as helicases and polymerases[1]. However, the individual binding events of proteins with NAs cannot be directly observed because they generally do not imply sufficiently large extension changes. Spatiotemporal resolution of single-molecule techniques is still making progress[2] and now allows observing the nanometric signals created by oligonucleotide hybridization (≥8 nucleotides (nt))[3] and helicase stepping[4]. However, the detection of simple binding events that involve Angström-scale extension changes below the second timescale is still out of reach. Coupling micromanipulation with Förster resonance energy transfer (FRET) fluorescence alleviates these lacks in spatiotemporal resolution[5]. Nevertheless, it is subject to standard fluorescence limitations, the most cumbersome being the need for substrate labeling and the uncertainty concerning the perturbation of the measurements by the fluorescent probe[6].

We report here a measurement scheme for single-molecule micromanipulation, called *kinetic locking*, that is neither based on extension changes nor on fluorescence. Detection and quantification are performed through the observation of the dynamics of a DNA fluctuating probe that is sensitive to changes in its direct chemical environment. The probe consists of a 10-bp DNA hairpin stretched at a force of 10 pN. At this force, the hairpin fluctuates between its duplex (closed) and single-stranded (ss, open) states[7]. The typical times between two successive changes of state are 10 ms. These fluctuations are highly sensitive to the applied force (a force increase of 0.7 pN doubles the opening rate of our probe), to the sequence of the hairpin[8], its apex loop length, and the presence of mismatches[9]. We also noticed that the kinetics is highly altered by the binding of another molecule. For example, a protein binding to ssDNA prevents the closing of the hairpin and will thus increase the time spent in this opened state. On the other hand, a protein binding to double-stranded DNA (dsDNA) prevents the opening of the hairpin and will consequently increase the time spent in its closed state. Therefore, these altered fluctuations can be used to detect the binding events.

We first validate our method by using a fluctuating probe, a short DNA hairpin, to measure the thermodynamics of hybridization of very short oligonucleotides (5, 6, and 7 bases). We show that our data are in good agreement with the nearest-neighbor (NN) prediction of the free energies[10].

In addition, we use the *kinetic locking* scheme to investigate DNA binding in the absence of ATP of the catalytic core of the *Escherichia coli* helicase RecQ, RecQ-ΔC. We demonstrate that our method enables the precise measurements of its binding kinetics and we show that the specific affinity of RecQ catalytic core replication forks (RFs) with a gap on the leading strand is a consequence of a much smaller leaving rate. We also show that the presence of RecQ stabilizes a fraying RF in its closed state, confirming its putative role of fork stabilizer[11].

## Results

### Design of an ultra-stable DNA fluctuating probe with magnetic tweezers. 
Our fluctuating probe consists of a 10-bp DNA hairpin that rapidly oscillates between its open and closed states (Fig. 1a, b). In order to constitute a sensitive probe, the transition rates of the system must display a strong temporal stability. For this particular application, we thus chose magnetic tweezers over AFM or optical tweezers[12] given their unrivaled stability in force and in temperature. Using the high-precision optical setup described in ref. [3], we were able to measure the transition rates of the 10-bp hairpin by tracking the 10-nm difference of extension between its two states. The time between two transitions is of the order of 10 ms, allowing us to measure $N = 10^4$ transitions during a typical acquisition time of 100 s and thus to keep the relative statistical error on the estimation of the kinetic rates below ≃5% during several hours. Figure 1c shows the temporal stability of these rates. The 7000 s of acquisition are separated into 20 chunks of 350 s. The kinetic rates inferred from each of them are all equal within a tolerance of 1 ms.

### Single-molecule detection of oligonucleotide hybridization with kinetic locking. 
Figure 2a shows the principle of *kinetic locking* applied to the characterization of the hybridization of oligonucleotides. Figure 2b shows how kinetic locking enables the direct visualization of the hybridization of a 7-base oligonucleotide. When the fluctuating DNA hairpin is stretched at 10 pN and no oligonucleotide is present in solution, the distribution of the times $T_{\text{open/closed}}$ spent in its open/closed states follows clearly a single-exponential distribution of parameters $k_f$ (folding rate) and $k_u$ (unfolding rate). Typical opening/closing times lie in the 10 ms range. Upon the injection of an oligonucleotide complementary to the hairpin sequence, the hairpin stays locked in the open state during ~250 ms (Fig. 2b, bottom). We interpret this time at the typical binding time of the oligonucleotide. This is confirmed by the change in the distribution of the open times. The latter evolves indeed from a single to double exponential (Fig. 2c), as predicted by the model underlying *kinetic locking* (Supplementary Note S1), and allows us to calculate the kinetic rates of binding of the 7-mer at ~9.5 pN (Supplementary Table S1).

### Kinetic locking allows the accurate measurement of the free energy of hybridization of oligonucleotides as short as 5 bases. 
We validate the accuracy of the method by performing single-molecule measurements of the thermodynamics of the hybridization of very short oligonucleotides (5, 6, and 7 bases). Indeed, our model predicts that the average time spent in the open state by the fluctuating probe should increase linearly with the concentration of oligonucleotides, the slope depending on the free energy of hybridization $\Delta G_{\text{binding}}$,

$$\overline{T_{\text{open}}}([\text{oligo}]) = \overline{T_{\text{open}}}(0)\left(1 + e^{-\frac{\Delta G^0_{\text{binding}}}{kT}}\frac{[\text{oligo}]}{c^0}\right), \quad (1)$$

where $c^0 = 1\,\text{M}$ in accordance with the standard definition of $\Delta G^0$. Interestingly, this dependence is also valid for the fast hybridization of 5- and 6-mer oligonucleotides whose typical binding times are smaller than the acquisition frequency (<10 ms) even if, in this case, $T_{\text{open}}$ follows a single-exponential distribution from which association and dissociation rates cannot be inferred separately (Supplementary Figure S1 and Supplementary Note S1, limiting case 2). Our experiments confirm the linear dependence of $\overline{T_{\text{open}}}$ with the concentration of oligomers (Figs. 2d and Supplementary Figure S2), while as expected the times spent in the closed state $\overline{T_{\text{close}}}$ do not depend on the concentration (Supplementary Figure S3). To control that this effect is not due to an increase of ionic strength due to the presence of charged NAs, we show that introducing an oligonucleotide that is not complementary to the sequence of the fluctuating hairpin (C–C mismatch in the third position) results in the absence of measurable change (Supplementary Figure S2, first panel). We compare in Table 1 the ΔG obtained with our measurements with the

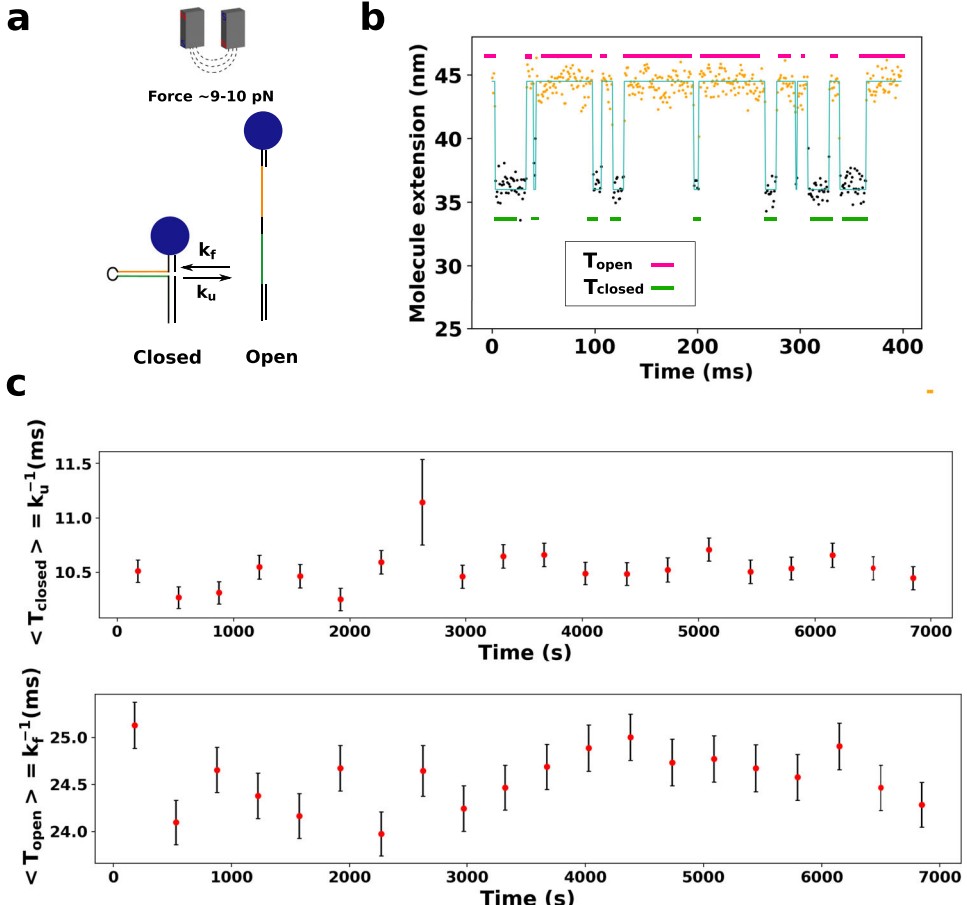

**Fig. 1 Kinetic rates of a DNA hairpin pulled with magnetic tweezers are stable during several hours. a** A force is applied to the hairpin with magnetic tweezers, allowing the latter to reach a state where it oscillates between its ssDNA and dsDNA configurations. **b** Measured extension of the hairpin as a function of time. Dots represent raw data. The blue line and the dot colors represent the inferred state. $T_{open}$ and $T_{closed}$ are defined as the time spent in the open and closed states by the hairpin. **c** Average $T_{closed}$ and $T_{open}$ measured during 7000 s of acquisition of the extension of a single molecule. The average is performed over time bins of 350 s. Error bars are calculated through bootstrap resampling.

widely used NN tables[10]. Usual salt corrections were used[13,14]. Our results are in good agreement with the data, thus validating our method. We also quantify the effect of adding an acridine group and a Cy3 group at the 5′ position of the 5-mer. The modified oligonucleotides display larger or equal hybridization energy than the 6-mer, confirming the strong interaction of the Cy3 group with DNA (Fig. 2e).

**Label-free measurement of the binding kinetics of the catalytic core of the *E. coli* RecQ helicase to an ssDNA substrate.** We finally use the *kinetic locking* scheme to investigate the DNA-binding kinetics of the catalytic core of the *E. coli* helicase RecQ, RecQ-ΔC (Fig. 1a of ref. [15]), in the absence of free nucleotides in solution. Figure 3 shows the evidence of the binding of RecQ-ΔC with DNA. The first probe (Fig. 3a, left panel) does not contain any ss gap when closed. When RecQ-ΔC binds on the ssDNA in the open state, it prevents the closing of the probe (Fig. 3a, middle panel). The distribution of times spent in the open state by the probe (Fig. 3a, right panel) allows us to determine the binding parameters of RecQ-ΔC in this configuration: $k_{off}^{NF} = 10.9 \pm 0.65\,\text{s}^{-1}$ and $k_{on}^{NF} = (1 \pm 0.08) \times 10^{-2}\,\text{nM}^{-1}\,\text{s}^{-1}$.

**The catalytic core of RecQ helicase stabilizes fraying replication forks with a gap on the leading strand.** RecQ is known to display a stronger affinity for substrates containing an RF with a gap

on the leading strand (LeGF)[16]. In order to assess the associated binding rates, we designed a slightly different fluctuating probe that contains a 14-nt ss gap on the leading strand in its closed state (Fig. 3b, left panel). In addition to blocking the substrate in its open state, RecQ also binds to the closed hairpin and maintains it in the closed state (Fig. 3b, middle panel and Fig. 3c). These binding events are longer, clearly distinguishable, and their frequency increases linearly with the concentration of RecQ-ΔC in solution (Fig. 3d). We found $k_{off}^{LeGF} = 0.9 \pm 0.15\,\text{s}^{-1}$ and $k_{on}^{LeGF} = (6.2 \pm 2.4) \times 10^{-2}\,\text{nM}^{-1}\,\text{s}^{-1}$. Figure 3e shows the 10-fold difference of dissociation rates between the configuration containing a fork with a leading gap (LeGF) and the configuration that does not (NF). $k_{off}$ is smaller in the case of the forked substrate. This confirms the larger affinity of RecQ for a forked substrate observed in bulk[16] with gel mobility shift assays, and allows us to quantify the associated binding rates. The fact that RecQ maintains the hairpin in its closed state corroborates structural studies showing that the cleft formed by the winged-helix (WH) and the $Zn^{2+}$ finger wraps around dsDNA[17,18]. It, furthermore, indicates that this cleft is strong enough to stabilize the fork hybridization while a force is applied. Since the force that we apply is directed against the stabilization of the fork, we checked whether the dissociation time of RecQ-ΔC was strongly affected by the force. In the narrow force range allowed by our experiment, we measured no effect of the force on the dissociation

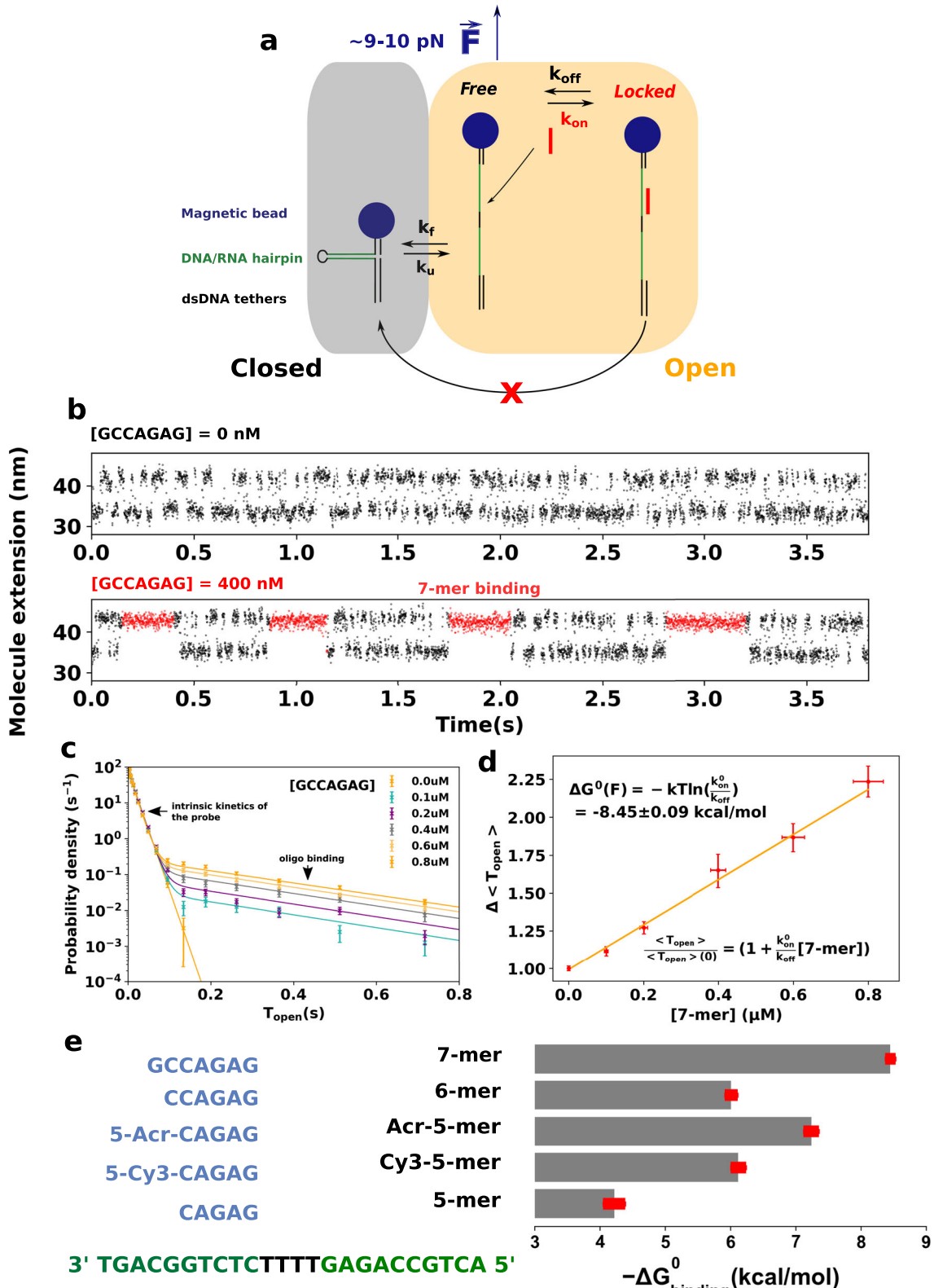

time RecQ-ΔC in this configuration (Supplementary Figure S4), while the typical time spent by the free hairpin in the closed state is divided by a factor 8. Interestingly, RecQ-ΔC also keeps the fork closed in the presence of a 7-base gap on the nascent lagging strand (Fig. 4), indicating that fork gripping by the enzyme is not dependent on the interaction with the free 5′ end.

## Discussion

The method that we report here allows the fluorescence-free detection of binding events of biochemical species with nucleic-acid substrates. We illustrated the temporal resolution of the method by performing single-molecule measurement of the thermodynamics of oligonucleotides as short as five bases. So far, temperature-jump

**Fig. 2 Measurement of the hybridization energy of short oligonucleotides with kinetic locking. a** Description of the assay. When an oligonucleotide binds to the probe, it blocks it transiently in the open state. **b** Larger time view of the measured extension of the hairpin in the absence of oligonucleotide in solution (top) and in the presence of the complementary 7-mer (bottom). **c** Distribution of the times spent by the DNA probe in its open state as a function of the concentration of 7-mer. The clear double-exponential distribution allows us to deduce the binding rates $k_{on}$ and $k_{off}$. **d** Relative evolution of the mean time spent by the DNA probe in the open state $\overline{T_{open}}$ as a function of the concentration of 7-mer. The slope allows us to compute the energy of binding. Y errors are computed through bootstrap resampling. X errors are based on a putative error of 5% on the concentration. Each data point consists of the averaging of at least 7000 events. **e** Free energies of binding of five different oligonucleotides measured with kinetic locking. The sequence of the probe is shown in green and the sequences of the oligonucleotides are in blue. Cy3 and acridine modifications increase substantially the binding energy. Errors are based on the covariance of the parameters inferred from the linear fits like the one shown in (**d**).

**Table 1 Comparison of the experimental binding energies (in kcal mol$^{-1}$) obtained by _kinetic locking_ with standard predictions by the nearest-neighbor model at 25 °C.**

| Oligomer | | | | | |
|---|---|---|---|---|---|
| | CAGAG | CCAGAG | GCCAGAG | 5′-MeO-Cl-Acr-CAGAG | 5′-Cy3-CAGAG |
| $\Delta G$—kinetic lock | −4.26 ± 0.08 | −6.01 ± 0.09 | −8.45 ± 0.09 | −7.24 ± 0.06 | −6.12 ± 0.09 |
| $\Delta G$—SantaLucia and Hicks | −4.05 | −6.01 | −8.42 | N/A | −5.57[a] |

_NA_ not available.
NN computations were made with the freely available software Biopython[35] after correction of two misreproduced values from the cited thermodynamic tables. $\Delta S°_{TA/AT} = -21.3$ e.u. and not −20.4 and $\Delta S°_{GG/CC} = -19.9$ and not −19.0. Displayed errors are statistical errors. The application of a force of 9.5 pN induces a systematic correction that lies between −0.015 and 0.040 kcal mol$^{-1}$ per nucleotide (see Supplementary Notes S2 and S3 and Supplementary Figures S6 and S7). All curves used to infer these values are shown in Supplementary Figure S2.
[a]The effect of the Cy3 group is estimated using the sequence-dependent values provided by Moreira et al. in ref. [6].

infrared spectroscopy is the only technique that was able to provide data[19] on the thermodynamics of ultrashort nucleotides. However, its accuracy is limited by the deuterium effects on hydrogen bonding and the short re-thermalization time $T_{th}$: in order to keep the association times smaller than $T_{th}$, nonbiological salt conditions must be used ([Na] ~ 200 mM and [Mg] ~ 40 mM). In addition to validating the method, the measurement of the hybridization thermodynamics of such short oligonucleotides is thus interesting per se. It will help to understand the role of specific sequence motifs on biological processes that involve transient hybridization such as silencing through RNA interference and to better understand unwanted off-target effects[20]. It will also help identify chemical modifications that stabilize short duplexes strongly enough so as to allow mechanical sequencing[21].

Most importantly, the technique allows the detection of discrete protein/NA binding events, alleviating the need for protein labeling while avoiding the disturbance of the measured interaction by an external probe. Label-free single-molecule measurements of the interaction of proteins with nucleic acids have been reported with nanohole optical tweezers[22]. The latter has the advantage of simplicity since the hairpin does not need to be tethered to a surface. However, they do not allow to access association times and remain qualitative since the force cannot be measured. On the other hand, the time resolution presented here (~10 ms) largely exceeds label-free single-molecule methods based on periodic force changes[23,24] that are well suited to study long dissociation times, but not short and repetitive bindings. Besides, this resolution could be further increased by using a high-speed camera to track the molecule extension (10 kHz) while concomitantly increase the folding rate of the probe by slightly decreasing the applied force. Compared to stopped-flow bulk assays, such as the one based on tryptophan fluorescence[25], it avoids photobleaching, hidden binding events, and the need to introduce chemical competitors to evaluate dissociation kinetics. Interestingly, the single-molecule nature of the experiment is susceptible to give important insights about the nature of the interaction between the protein and its substrate, as was shown by the unexpected stabilization of the probe by RecQ catalytic core. The main limitation is the narrow force range (~8–15 pN), since

the NA substrate needs to be stretched in order to induce its fluctuations. However, this corresponds to the typical stall forces of NA processing motors during critical processes such as transcription[26], translation[27], or replication[28] and is thus consistent with physiological conditions. Our assay can be applied to a broad spectrum of DNA/RNA-binding proteins (notably transcription factors, polymerases, helicases, primases), and used to study their sequence and substrate specificity. Another potential application is the precise quantification of the inhibition of enzyme/NA binding by potential antiviral drugs, in a context where replicative enzymes are increasingly considered as relevant drug targets (helicase-primase of Herpes[29], helicase nsp13 of SARS-CoV-2[30]). By providing reliable single-molecule measurements while avoiding the functional interference of labeling groups, _kinetic locking_ will complement the toolbox of enzymologists for the kinetic characterization of NA/protein complexes.

## Methods

**Design of stable fluctuating probe**. The fluctuating probe consists of a 10-bp DNA hairpin that rapidly oscillates between its open and close states. Using the high-precision optical setup described in ref. [3], we measure the transition rates of the 10-bp hairpin by tracking the 10-nm difference of extension between its two states. The time between two transitions is of the order of 10 ms, allowing us to measure $N = 10^4$ transitions during a typical acquisition time of 100 s and thus to reduce the relative error made on the estimation of the kinetic rates to $\sqrt{N} \simeq 1\%$. Figure 1b shows the temporal stability of these rates. The 7000 s of acquisition are separated into 20 chunks of 350 s. The kinetic rates inferred from each of them are all equal within a tolerance of 1 ms.

**Bead preparation**. All single-molecule experiments are performed at 25 °C. All the DNA substrates used in the presented assays are synthesized ss oligonucleotides. Their sequences are indicated in Supplementary Table S2. Their 5′ end is complementary to a 57-base 3′ dibenzocyclooctyl modified long oligonucleotide (Oli1) that is attached to azide-functionalized surfaces (PolyAn 2D Azide) through a 2-h-long incubation (100 nM Oli1, 500 m NaCl). Their 3′ end is complementary to a 58-base oligonucleotide (Oli2). Oli2 contains two biotin modifications at its 5′ end. The ssDNA substrate is first hybridized with Oli2 by mixing both oligos at 100 nM in 100 mM NaCl, 30 mM, Tris-HCl pH 7.6. 5 μL of streptavidin-coated Dynabeads MyOne T1 (Thermo Fisher) are washed three times in 200 μL of passivation buffer (140 mM NaCl, 3 mM KCl, 10 mM Na$_2$HPO$_4$, 1.76 mM KH$_2$PO$_4$, bovine serum albumin 2%, Pluronic F-127 2%, 5 mM EDTA, 10 mM NaN$_3$, pH 7.4). The result of the hybridization between the substrate and Oli2 is diluted down to 2 pM and then incubated for 10 min with the beads in a total volume of 20 μL of passivation

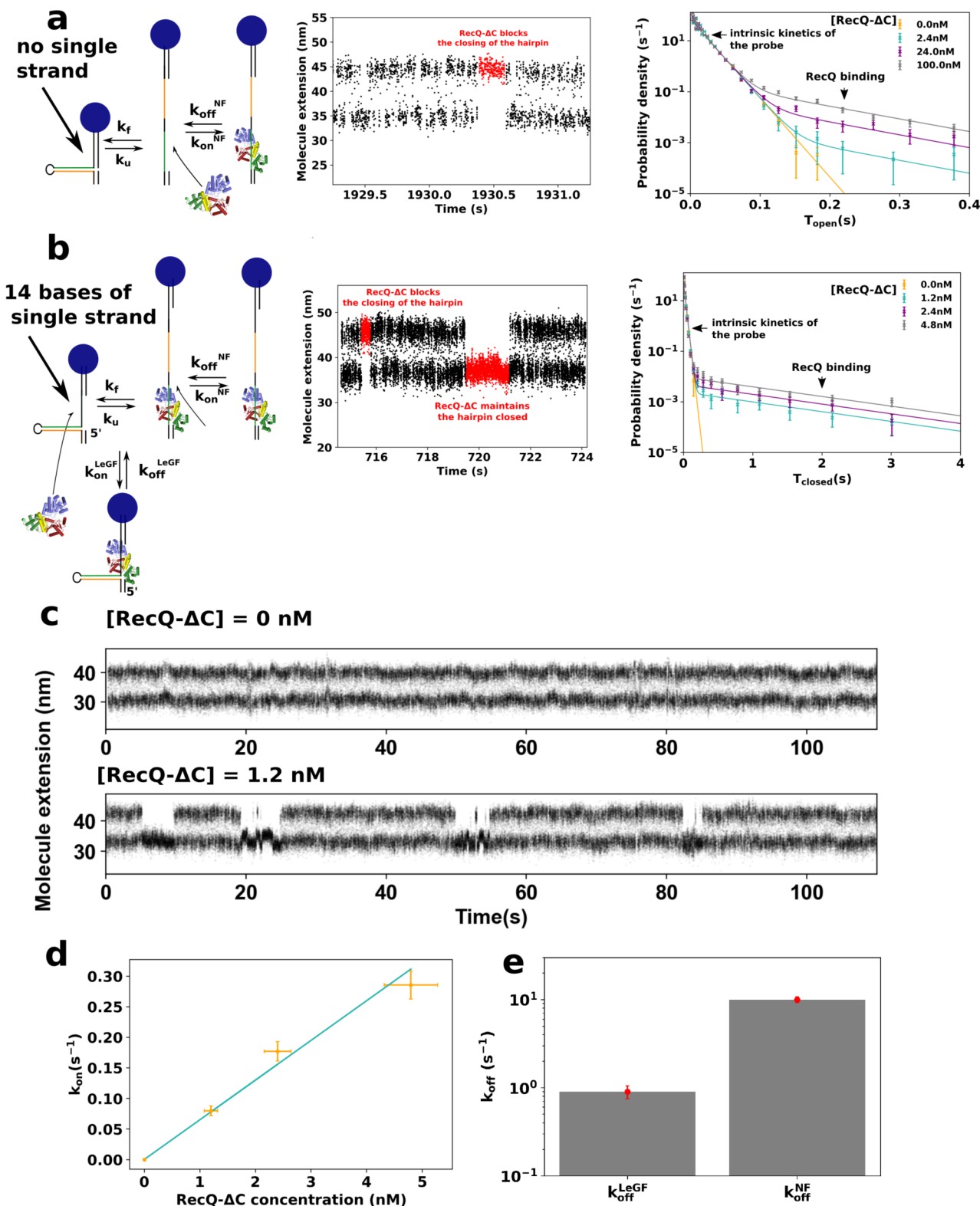

buffer. The beads are then rinsed three times with a passivation buffer in order to remove unbound DNA. One microliter of the bead solution is then introduced in the cell coated with Oli1 and filled with passivation buffer, and finally incubated for 5 min. Excess unbound beads are washed out by flowing passivation buffer.

**Sample preparation**. For the DNA probes, three different constructs (HP1/HP3/HP2) are used, which sequences are given in Supplementary Table S2. They consist

of synthesized ssDNA of length, respectively, 160, 153, and 139 bases. Once hybridized to Oli1 and Oli2, the size of the remaining ssDNA reduces to (45/38/24) bases. Twenty-four of these bases fold into a 10 bp hairpin with a 4-base apex T-loop.

The fluctuating probe used to measure oligonucleotide binding is HP1. Once the hairpin is hybridized, 7 ss bases are left on the 5′ side and 14 on the 3′ side. Measurements are performed in hybridization buffer (20 mM Tris-HCl pH 7.6, 3 mM MgCl$_2$, 100 mM KCl). The oligonucleotides are ordered dry (Eurogentec)

**Fig. 3 Detection and quantification of the binding of RecQ-ΔC with various DNA substrates using kinetic locking. a** A 10-bp hairpin without ssDNA gap (left) is used as a fluctuating probe. (Middle) The binding of RecQ-ΔC can be observed through the transient blocking of the hairpin (HP) in its open state. (Right) Distribution of times spent by the hairpin in its open state as a function of RecQ-ΔC concentration, fitted by the double-exponential law predicted by the kinetic model (Supplementary Note S1). Relative errors of a given bin are taken as $\sqrt{N}$, where $N$ is the number of points in the bin. **b** A 10-bp hairpin with a 14-nt ssDNA gap (left) is used as a fluctuating probe, simulating a leading-gapped replication fork, LeGF (middle). The binding of RecQ-ΔC on the ssDNA gap results in the transient stabilization of the hybridized state of the HP. (Right) Distribution of times spent by the hairpin in its closed state as a function of RecQ-ΔC concentration, fitted by a double-exponential law. **c** Larger time view of probe **b** with and without RecQ-ΔC. **d** Association rate $k_{on}$ in the closed state for configuration **b** as a function of RecQ-ΔC concentration. Y relative errors correspond to the inverse of the square roots of the number of observed events. X errors are based on a 10% relative error on the concentration. **e** Comparison of the dissociation rates $k_{off}$ of RecQ-ΔC with ($k^{LeGF}$) and without ($k^{NF}$) the presence of an ssDNA gap. Error bars are derived from bootstrap resampling (see "Methods").

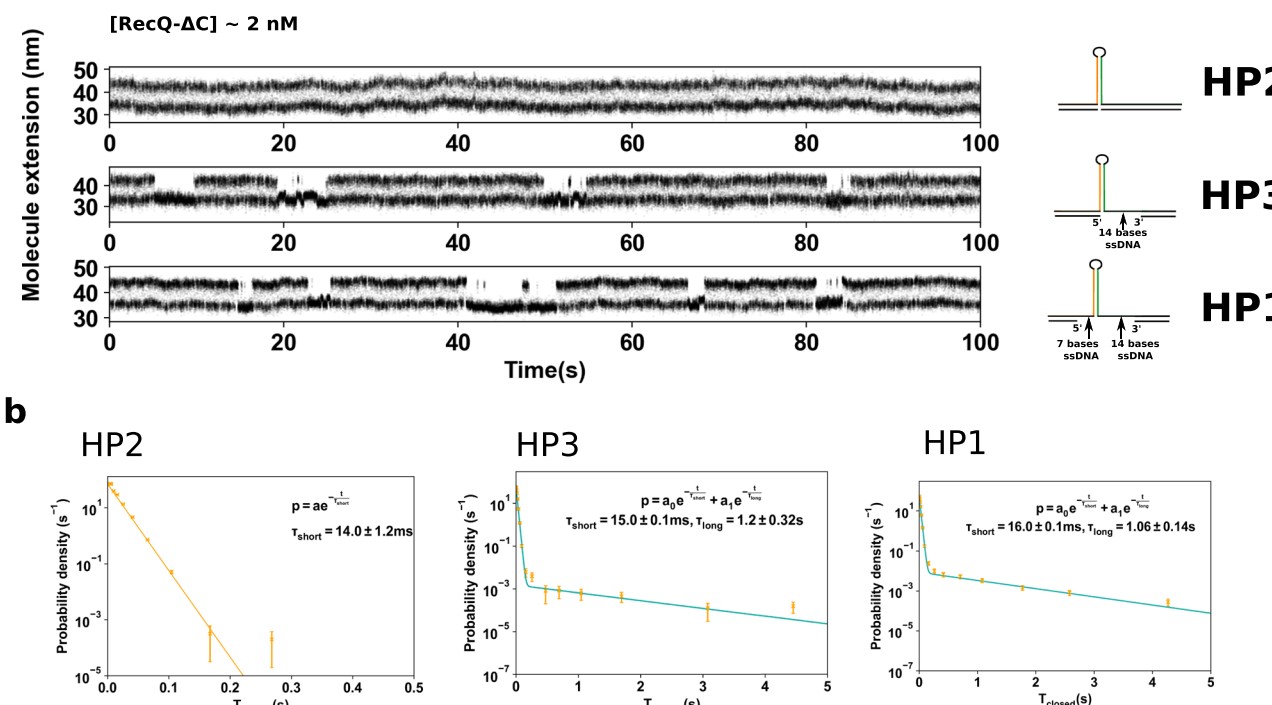

**Fig. 4 Stabilization of a fraying DNA fork by RecQ-ΔC as a function of the configuration of the nascent strands. a** Time-course experiments showing the extension changes of the fluctuating probe in presence of RecQ-ΔC at $2 \pm 1$ nM for different probes (right) whose sequence can be found in Table S2. The stabilization of the closed state only happens when there is a single-stranded gap on the leading strand (HP1 and HP3), and is happening regardless of the presence of a single-stranded gap on the lagging strand (HP1). **b** Distributions of the time spent in the closed states for the three hairpins.

and diluted in the same buffer preparation as the one used during the hybridization assay in order to avoid any change in salinity upon injection. Concentrations are calculated based on the quantity indicated by the provider.

RecQ-ΔC is purified as described in ref. [15] and is diluted without ATP in RecQ buffer at a concentration specified in the main text. The fluctuating probes HP2 (without 3′ ssDNA gap) and HP3 (with 3′ ssDNA gap) are used. Measurements are performed in RecQ buffer (20 mM Tris-HCl pH 7.6, 3 mM MgCl₂, 50 mM NaCl). The concentration of the enzyme is assessed through ultraviolet absorbance (210–340 nm) with a Nanodrop (Thermo Fisher) spectrophotometer ($\epsilon = 45,840$ M$^{-1}$ cm$^{-1}$).

**Data acquisition**. The extension of the fluctuating probe is acquired in real time using the high-resolution magnetic tweezers setup described in ref. [3] at an acquisition frequency of 1300 Hz with a CMOS camera (UI-3060CP-M, IDS Ueye). The home-made C/C++ acquisition program *Xvin* can be found online[31].

**Statistics and reproducibility**. The changes of states (open/close) of the fluctuating probe are detected automatically with the program *Xvin*, when the summed change of extension over a monotonous section of the extension curve is larger than 8 nm.

Times spent in the open state, $T_{open}$, correspond to the difference between the detected times of two successive opening and closing events. Times spent in the closed state, $T_{closed}$, correspond to the difference between the detected times of two successive closing and opening events.

Average times $\overline{T_{open/closed}}$ correspond to naive mean values. Errors are computed using bootstrap resampling. All averages of the hairpin closing and opening times are performed on samples containing at least 4000 independent events. The exact number of events for each experiment is shown in Supplementary Figure S5.

The computation of the probability is performed in the following way. Individual open times are partitioned into N bins of size $s_i$ whose size increase geometrically ($s_{i+1} = s_i \times a$, where $a \in [1.2, 2]$). This partitioning gives rise to count number $n_i$. The number of events $n_i$ in each bin is divided by the size of the bins $s_i$. They are finally normalized by dividing by $\sum_i s_i n_i$ so that the integral of the probability density equals 1.

In the case of concentration-dependent double-exponential distributions, the times are fitted to the exact formula provided in Supplementary Note S1 with the SciPy[32] curve-fit function. A single fit is used for all concentrations. Errors on the fit are computed based on bootstrap resampling: 100 samples of the same size as the original data sample are generated by resampling the original data with replacement. The fitting procedure is then applied to all samples, returning a

distribution of inferred kinetic parameters. The values shown in the article correspond to the averages of these distributions while the errors correspond to their standard deviations. The proportion of long times in the double-exponential distribution is then verified with a binning-free expectation maximization (EM) algorithm based on the Pomegranate[33] Python module.

In order to compute the association rate $k_{on}^{LeGF}$ of RecQ-ΔC to the ssDNA gap located outside the fork, the number of long times in the double-exponential distribution is measured as described above and are divided by the whole time spent by the probe in the unbound state of the hairpin. In this case, the time spent in both the closed and the open state is considered since the ss binding site is available in both states, contrarily to the case of the association of oligonucleotides or of the association of RecQ-ΔC on the fluctuating portion of the hairpin $k_{on}^{NF}$, where the enzyme and the DNA oligonucleotide can only bind in the open state.

**Reporting summary**. Further information on research design is available in the Nature Research Reporting Summary linked to this article.

## Data availability
Numeric data supporting all graphs and charts are given in Supplementary Data 1. The lists of times spent in the open and closed states of the hairpin that were used to infer the kinetic and thermodynamic constants in this study are available in Supplementary Data 2. All raw time traces generated during the current study are available from the corresponding author on reasonable request. All data analysis procedures are available online[34].

## Code availability
The source code of the acquisition program is available online[31].

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

## Acknowledgements
We wish to acknowledge the engineering support of José Da Silva Quintas. We also acknowledge stimulating discussions with Phong Lan Thao Tran, Fatima Hamouri, David Bensimon, and Jimmy Ouellet. This study was supported by the ANR CLEANMD grant (ANR-14-CE10-0014) and ANR G4-CRASH (ANR-19-CE11-0021-01) from the French Agence Nationale de la Recherche (to V.C.) and by the European Research Council grant Magreps [267 862] (to V.C.). Work in the group of V.C. is part of "Institut Pierre-Gilles de Gennes" ("Investissements d'Avenir" program ANR-10-IDEX-0001-02 PSL and ANR-10-LABX-31) and the Qlife Institute of Convergence (Université PSL).

## Author contributions
M.R. conceived, performed, and analyzed the experiments. M.R. wrote the manuscript. M.R., V.C., J.V.-O., and J.-F.A. discussed the results. B.D. purified RecQ-ΔC. V.C. and J.-F.A. built the magnetic tweezers and supervised the research. All authors reviewed the manuscript.

## Competing interests
The authors declare no competing interests.
