## [Transparent Peer Review File · Communications Biology]

Reviewers' comments:

Reviewer #2 (Remarks to the Author):

In the manuscript "Fluorescence-free quantification of protein/nucleic-acid binding through single-molecule kinetic locking" the authors use magnetic beads to probe unzipping of DNA and thereby infer the interaction with a protein to a single strand of the DNA. The method seems very similar to the one reported in this work:

<https://doi.org/10.1364/BOE.5.001886>

The similarities are:

- probing 10 bp DNA hairpin interaction
- probing interaction with single proteins
- fluorescent label free
- binding potentials are inferred from transition times
- single molecule approach

These similarities overlap substantially with what the authors claim is essentially new about their approach.

While authors recognize the use of optical tweezers to do essentially the same things as they report (with reference to [3]), they seem to be unaware of the work cited above. Due to the distinct similarities with this past work, it is recommended that major revisions are presented to provide context of that past work in the present findings.

Reviewer #3 (Remarks to the Author):

Rieu et al. present a novel magnetic tweezers based single-molecule manipulation method for direct detection of biomolecular binding without use of fluorescent probes. The single molecule method measures binding parameters such as k_{on} and k_{off} and free energy of interaction. This method of measurement is based on a kinetic locking mechanism that relies on the changes in the lifetimes and distribution of open/close states of a DNA hairpin probe to detect a binding event in real time. To demonstrate the suitability of the method, the authors measure annealing of short complementary DNA sequences and binding of the catalytic core of *Escherichia coli* RecQ helicase to a forked structure in the hairpin DNA probe. The changes in T_{open} and T_{closed} provides the necessary data to determine the k_{on} and k_{off} and the energy of interaction. This method will complement other methods in studies of DNA-DNA and DNA-protein interactions such as ensemble stopped-flow, single molecule FRET or fluorescence with the advantage that the protein or nucleic acid does not need to be labeled.

The reported method is novel, the data are convincing, and the experimental and analytical methods are described clearly. I have a few queries, which I would like the authors to respond to:

1. Figure 1 presents data on annealing of a short oligo to the fluctuating hairpin. The panels are fairly clear, except for panel D, where probability density (s^{-1}) is plotted against T_{open} (s). The authors need to provide a more detailed description of how the Y-axis which is probability density was obtained. The figure legend needs to clearly describe panel D and E.

2. A detailed description in the supplementary information should be added for how k_{on} and k_{off} values are obtained for the benefit of the general reader.

1. Section S.I derivations should be more descriptive for general audience. The authors should carefully define the parameters used in the equations and add missing steps in the derivations. For example, what C_0 represent in the equation has not been made clear in the text.

3. The authors discuss the possibility that the binding properties of the complementary DNA oligonucleotide could be affected by the force applied to the hairpin DNA probe. Please address how the force impacts protein-binding to the hairpin DNA probe. Are the k_{on} and k_{off} values affected by

the force?

4. Hishida et al. have demonstrated that RecQ binds to double stranded DNA as well as a DNA fork with no gap (such as the HP2 probe used in this study), albeit with low binding affinities (Reference no. 19). It is expected that high concentrations of RecQ could detect binding events on 'HP2' probe. Can the authors comment on the range of protein-DNA interactions possibly studied with this method in terms of the binding affinities?

5. It is surprising that RecQ stabilizes the closed state, when I would have expected that RecQ being a helicase would destabilize the fork junction and hence destabilize the closed state. Is this because the experiments were carried out in the absence of ATP or ATP analog? The authors should comment further on this unusual observation.

6. The time-course data for RecQ (at ~2 nM concentration) binding with hairpin probes HP1, HP2, and HP3 are presented in Fig. S4. It would be interesting to see the comparison of probability distribution data in the three cases.

Minor points:

2. Correct E. coli in abstract and on Page 2 to Escherichia coli/E. coli.

3. The color scheme in Figure 1A: The hairpin forming region, including the TTTT loop, is represented in green in the closed configuration. In the open state though, there is a black segment in the middle, probably representing the loop. If so, the loop region in the closed state should be changed to black color.

4. Page 1, last paragraph: "...the distribution of the times Topen/close spent in its closed/open states follow clear single-exponential distributions...". It be "...the times Topen/close spent in its open/close states..."?

Response to reviewers

Reviewer 2:

In the manuscript "Fluorescence-free quantification of protein/nucleic-acid binding through single-molecule kinetic locking" the authors use magnetic beads to probe unzipping of DNA and thereby infer the interaction with a protein to a single strand of the DNA. The method seems very similar to the one reported in this work:

<https://doi.org/10.1364/BOE.5.001886>

The similarities are:

- probing 10 bp DNA hairpin interaction
- probing interaction with single proteins
- fluorescent label free
- binding potentials are inferred from transition times
- single molecule approach

These similarities overlap substantially with what the authors claim is essentially new about their approach.

We thank the reviewer for pointing out this technique but we strongly disagree regarding the overlap with our method. We are aware of this type of measurement of the interaction of a hairpin with a protein, which can be found in an earlier work that we refer to in our manuscript (reference 23). However:

- The quantitative nature of the method may be questioned since the extent of the destabilization of the hairpin by the nano-hole tweezers cannot be known nor controlled. As the authors mention, the force is dependent on the system that is observed "*The optical forces are modified by the trapped DNA molecule and scale with the polarizability of the DNA fragment*". The fact that the authors of this article can not measure a difference of unfolding times between a 10 bases and a 12 bases hairpin shows the limit of the quantification permitted by the method.
- This type of method is restricted to the case of proteins stabilizing the duplex structure and not to proteins binding to single-stranded nucleic acids.
- It does not provide a measure of the association times but can only access dissociation times. In particular, this explains why the method does not allow to retrieve the binding energy nor the dissociation constant K_d of the protein. Only a change of energy of the transition state can be estimated through the use of the dissociation time.

Contrarily to the above mentioned technique, *our manuscript* reports a *new, quantitative and general* method allowing us to assess binding and unbinding kinetics with a wider range of substrates, including ssDNA, and that also allows measuring association times. Furthermore, our experiments with oligonucleotides demonstrate an unprecedented precision for the single-molecule measurement of the thermodynamics of association and dissociation of such short sequences. As stated and explained in the manuscript, the precision is obtained through:

- A thorough control of the force and of the temperature, which allows accurate measurements of kinetic and thermodynamics.
- The sampling of a high number of events (typically several thousands of hairpin fluctuations and a few hundreds of protein binding events in a few minutes) obtained through the use of a fluctuating probe. In comparison with the paper quoted by the reviewer the number N of events is not stated, error bars are not provided, and analyzing the cumulative histogram reveals that N is in the range of 10.

While authors recognize the use of optical tweezers to do essentially the same things as they report (with reference to [3]), they seem to be unaware of the work cited above.

We do not understand this remark. The article that the reviewer refers to is a paper that we recently published and that describes a new optical detection method. In particular this method allowed to reach the time resolution needed to perform this work. As a consequence we are very well aware of its content. First, it does not make use of optical tweezers but of magnetic tweezers. Second, and more importantly concerning the novelty of the submitted work it does not introduce the kinetic locking, the method presented here that allows us to detect binding events that do not involve extension changes. Concerning "the work cited above" we already gave our arguments.

Due to the distinct similarities with this past work, it is recommended that major revisions are presented to provide context of that past work in the present findings.

As we argued the present work allows measurements that are far beyond the scope of the references given by reviewer 2. We do have the impression that the sentence : "However the detection of simple binding events that involve Ångström-scale extension changes below the second timescale are still out of reach" makes clear in which context the kinetic locking can bring new information.

Reviewer 3:

We thank reviewer 3 for its thorough reading of our work, its understanding of our technique and its relevant remarks.

Rieu et al. present a novel magnetic tweezers based single-molecule manipulation method for direct detection of biomolecular binding without use of fluorescent probes. The single molecule method measures binding parameters such as k_{on} and k_{off} and free energy of interaction. This method of measurement is based on a kinetic locking mechanism that relies on the changes in the lifetimes and distribution of open/close states of a DNA hairpin probe to detect a binding event in real time. To demonstrate the suitability of the method, the authors measure annealing of short complementary DNA sequences and binding of the catalytic core of Escherichia coli RecQ- Δ C helicase to a forked structure in the hairpin DNA probe. The changes in T_{open} and T_{closed} provide the necessary data to determine the k_{on} and k_{off} and the energy of interaction. This method will complement other methods in studies of DNA-DNA and DNA-protein interactions such as ensemble stopped-flow, single molecule FRET or fluorescence with the advantage that the protein or nucleic acid does not need to be labeled. The reported method is novel, the data are convincing, and the experimental and analytical methods are described clearly. I have a few queries, which I would like the authors to respond to:

- 1. Figure 1 presents data on annealing of a short oligo to the fluctuating hairpin. The panels are fairly clear, except for panel D, where probability density (s^{-1}) is plotted against T_{open} (s). The authors need to provide a more detailed description of how the Y-axis which is probability density was obtained. The figure legend needs to clearly describe panel D and E.
- We have added the following explanation in the methods. The computation of the probability is performed in the following way. Individual open times are partitioned into N bins of size s_i whose size increases geometrically ($s_{i+1} = s_i \times a$, where $a \in [1.2, 2]$). This partitioning gives rise to counts number n_i . The number of events n_i in each bin is divided by the size of the bins s_i . They are finally normalized by dividing by $\sum_i s_i n_i$ so that the integral of the probability density equals 1. The use of bins with increasing size is made necessary by the existence of two typical times that are of different orders of magnitudes.
- 2. A detailed description in the supplementary information should be added for how k_{on} and k_{off} values are obtained for the benefit of the general reader.
- Following the reviewer's advice, we detailed the description of the inference of the parameters in the section Data analysis of the methods. In particular, we detail a subtlety regarding the computation of k_{on} that we had omitted to mention in the first version. In the case of oligonucleotides binding to RecQ- Δ C in the open fork, the binding can only occur in the open state, and thus k_{on} corresponds to the number of binding events divided by the whole time spent in the open free state. In the case of RecQ- Δ C to the single-stranded DNA gap, the binding can occur in the open state and in the closed state, and thus k_{on} corresponds to the number of binding events divided by the whole time spent by the probe in the free state (closed and open).
- 3. Section S.I derivations should be more descriptive for general audience. The authors should carefully define the parameters used in the equations and add missing steps in the derivations. For example, what C_0 represent in the equation has not been made clear in the text.
- We detailed the definitions of each variable in section S.I., and we have added the definition of C_0 in the main text.
- 4. The authors discuss the possibility that the binding properties of the complementary DNA oligonucleotide could be affected by the force applied to the hairpin DNA probe. Please address how the force impacts protein-binding to the hairpin DNA probe. Are the k_{on} and k_{off} values affected by the force?
- Binding properties are indeed functions of the force. As discussed in the conclusion, the force range where the experiments can be performed is quite narrow, given that it should allow the fluctuations of the probe between the open and the closed states. This narrow range presents, however, an advantage compared to standard magnetic tweezers experiments since it allows a highly reproducible force between experiments, avoiding the uncertainty related to the magnetization of the bead.

In the case of oligonucleotides, theory provides an estimation of the impact of the force as shown in the supplementary S. V. In the case of protein binding, such a theory is absent, making difficult a thorough extrapolation of the data to different forces. However, given that RecQ- Δ C is involved in replication and that the stalling force of a DNA polymerase is typically of few tens of pN, evaluating kinetic and thermodynamic data at $10pN$ seems to be as relevant as at the usual zero-force bulk experiments.

We could expect the force to play an important role on the binding on the closed state, since RecQ- Δ C holds the hairpin closed. Following the remark of reviewer 2, we thus measured the impact of the force on the dissociation of RecQ- Δ C on a fork in the force range allowed by our experiment (8-10 pN). The results are shown in Figure S5 in the manuscript and reproduced below. While the dynamics of the free hairpin critically depends on the force, there is no significant change of the dissociation time of RecQ- Δ C in this force range. We cannot completely exclude a change at lower force, but our results suggest that this dependence is limited. We added this discussion in the manuscript.

Figure 1. Force dependence of the binding of RecQ- Δ C in the closed state of the hairpin. The short times (green) correspond to the times spent in the closed state by the freely fluctuating hairpin. These times are strongly dependent on the force and are used to infer the force. The long times (orange) correspond to the binding times of RecQ- Δ C.

- 5. Hishida et al. have demonstrated that RecQ- Δ C binds to double stranded DNA as well as a DNA fork with no gap (such as the HP2 probe used in this study), albeit with low binding affinities (Reference no. 19). It is expected that high concentrations of RecQ- Δ C could detect binding events on ‘HP2’ probe. Can the authors comment on the range of protein-DNA interactions possibly studied with this method in terms of the binding affinities?

As you suggested, we analyzed the time spent in the closed state in the case of "HP2", and we realized that we had missed a small effect on the time spent in the closed state, indicating a binding of RecQ in the closed state even in the absence of a single-stranded DNA gap, but this time with a much lower affinity. The association constant being significantly smaller than on the open state, we have fewer points in our data and the statistics are weaker (see figure 2). Still, we find $k_{\text{on}}^{\text{HP2,closed}} = 1 \pm 0.3 \times 10^{-3} \text{ nM}^{-1} \text{ s}^{-1}$, more than one order of magnitude smaller than the $k_{\text{on}} = (6.2 \pm 2.4) \times 10^{-2} \text{ nM}^{-1} \text{ s}^{-1}$ in the presence of a single-stranded gap. On the other hand, $k_{\text{off}}^{\text{closed,HP2}} = 5 \pm 0.9 \text{ s}^{-1}$, significantly larger than $k_{\text{off}}^{\text{LeGF}} = 0.9 \pm 0.15 \text{ s}^{-1}$ in the presence of a single-stranded gap. This indicates that the dissociation constant from the closed fork in HP2 of $4 \pm 0.9 \mu\text{M}$, to be compared to $15 \pm 3 \text{ nM}$ in a presence of a single-stranded gap (HP3). While the dissociation constant for HP3 is very close to the one found by Hishida in bulk (13 nM), the affinity that we observe in the case of HP2 ($4 \mu\text{M}$) is much weaker than the one reported by Hishida ($K_d = 130 \text{ nM}$). It could come from the fact that Hishida et al. study WT-RecQ while we only study the catalytic core in our proof-of-principle. However, we cannot exclude a strong imprecision of the bulk measurements in the study of Hishida et al.: the change of electrophoretic behavior of the complex is indeed very abrupt in the case of substrate F and the measurement of K_d basically relies on two points.

Figure 2. Distribution of the times spent in the closed state with HP2 (no gap).

Figure 3. Reproduction of figure 1 from Hishida *et al.*. [recQ-wildtype] (from left to right) 1, 5, 10, 50, 100, 500, 1000 nM. [DNA] = 10 nM.

- 6. It is surprising that RecQ- Δ C stabilizes the closed state, when I would have expected that RecQ- Δ C being a helicase would destabilize the fork junction and hence destabilize the closed state. Is this because the experiments were carried out in the absence of ATP or ATP analog? The authors should comment further on this unusual observation.
- The reviewer's remark is interesting. We studied the unwinding activity of RecQ- Δ C in previous work and focused here on its binding without ATP. We show that helicase without ATP has a high affinity for a fork substrate. This property seems essential for the enzyme to find its template. Having a high affinity for a NA fork is likely to involve a binding with both strands and thus a stabilizing effect. On the other hand, structural studies (ref 21 and 22 of our manuscript), suggest that this affinity for the fork is caused by the cleft formed by the winged helix and the Zn²⁺ finger. These two domains do not contain the motor domains of RecQ- Δ C responsible for ATPase activity and translocation. Thus, the fact that RecQ- Δ C holds the fork, preventing its fraying, does not seem incompatible with its translocation and unwinding activity. We made some experiments with ATP that may support our claim. On the figure below, kinetic locking of

RecQ- Δ C is performed with ATP on the hairpin HP3. The figure shows in red an event of active unwinding of the fork by RecQ- Δ C. Interestingly, while RecQ- Δ C unwinds the hairpin in a few steps, the frequency of the hairpin fluctuation obviously decreases compared to the free state of the hairpin (in black). **For us, this means that RecQ- Δ C holds both strands of the fork while translocating and unwinding the hairpin.**

Figure 4. Event of active unwinding of HP3 by RecQ- Δ C in saturating conditions of ATP (1 mM).

We did not add this data to the manuscript because we only recorded a few tens of events of active unwinding, which is not enough to present rigorous statistics. Gathering more data would imply producing a new batch of more protein, which is time consuming, to study this question which seems a little bit out of the scope of this paper.

- 7. The time-course data for RecQ- Δ C (at 2 nM concentration) binding with hairpin probes HP1, HP2, and HP3 are presented in Fig. S4. It would be interesting to see the comparison of probability distribution data in the three cases. Following the reviewer’s advice, we added the time distributions in the figure S4.

Minor points:

- 8. Correct E. coli in abstract and on Page 2 to Escherichia coli/E. coli. We corrected that.
- 9. The color scheme in Figure 1A: The hairpin forming region, including the TTTT loop, is represented in green in the closed configuration. In the open state though, there is a black segment in the middle, probably representing the loop. If so, the loop region in the closed state should be changed to black color. Done
- 10. Page 1, last paragraph: “... the distribution of the times $T_{open/close}$ spent in its closed/open states follow clear single-exponential distributions...”. It be “... the times $T_{open/close}$ spent in its open/close states...”? Yes, thank you for pointing out this typo.

REVIEWERS' COMMENTS:

Reviewer #2 (Remarks to the Author):

The authors disagree with my evaluation that the work they presented has great similarity with a previous work due to:

"

- probing 10 bp DNA hairpin interaction
- probing interaction with single proteins
- fluorescent label free
- binding potentials are inferred from transition times
- single molecule approach

"

Instead of addressing these similarities (which are definitely true), they talk about reference 23, which does not use such short fragments (10 bp) and requires tethers, and they criticize the work as being not quantifiable, not generalizable, and not quantifying "on" times. These three criticisms are not related to the similarities to their work, and I also believe them to be false, but this is a greater discussion that is not related to the clear similarities, so I will not go on that tangent. The authors assert that they have produced the first technique to have those characteristics, which is false based on the previous work I referenced. If they truly believed that their technique was distinct from that previous work, they would reference that work and explain how it is distinct. Instead, they expand only in the rebuttal letter (closed communication) and they do not address the similarities, that they are claiming are unique to their work.

Therefore, I believe that this work presents a technique that is very similar to past works and should not be published when it fails to acknowledge previous highly similar works.

Reviewer #3 (Remarks to the Author):

The authors have addressed all my questions, the manuscript should be published without delay